# Ovocystatin Induced Changes in Expression of Alzheimer’s Disease Relevant Proteins in APP/PS1 Transgenic Mice

**DOI:** 10.3390/jcm11092372

**Published:** 2022-04-23

**Authors:** Bartlomiej Stanczykiewicz, Jakub Gburek, Maria Rutkowska, Marta Lemieszewska, Krzysztof Gołąb, Katarzyna Juszczyńska, Aleksandra Piotrowska, Tadeusz Trziszka, Piotr Dzięgiel, Marzenna Podhorska-Okołów, Agnieszka Zabłocka, Joanna Rymaszewska

**Affiliations:** 1Department of Psychiatry, Division of Consultation Psychiatry and Neuroscience, Wroclaw Medical University, 50-367 Wroclaw, Poland; m.lemieszewska@gmail.com (M.L.); joanna.rymaszewska@umw.edu.pl (J.R.); 2Department of Pharmaceutical Biochemistry, Wroclaw Medical University, 50-556 Wroclaw, Poland; jakub.gburek@umw.edu.pl (J.G.); krzysztof.golab@umw.edu.pl (K.G.); katarzyna.juszczynska@umw.edu.pl (K.J.); 3Department of Pharmacology, Wroclaw Medical University, 50-345 Wroclaw, Poland; maria.rutkowska@umw.edu.pl; 4Department of Human Morphology and Embryology, Division of Histology and Embryology, Wroclaw Medical University, 50-368 Wroclaw, Poland; aleksandra.piotrowska@umw.edu.pl (A.P.); piotr.dziegiel@umw.edu.pl (P.D.); 5Department of Animal Products Technology and Quality Management, Wroclaw University of Environmental and Life Sciences, 51-630 Wroclaw, Poland; tadeusz.trziszka@upwr.edu.pl; 6Department of Human Biology, Faculty of Physiotherapy, University School of Physical Education, 51-612 Wroclaw, Poland; 7Division of Ultrastructure Research, Wroclaw Medical University, 50-368 Wroclaw, Poland; marzenna.podhorska-okolow@umw.edu.pl; 8Department of Microbiology, Hirszfeld Institute of Immunology and Experimental Therapy, Polish Academy of Sciences, 53-114 Wroclaw, Poland; agnieszka.zablocka@hirszfeld.pl

**Keywords:** ovocystatin, chicken cystatin, cystatin C, Alzheimer’s disease, mice

## Abstract

Background: Ovocystatin is marked by structural and biological similarities to human cystatin C, which plays an important role in the course of neurodegenerative diseases. Recently, it has been shown that ovocystatin might prevent aging-related cognitive impairment in rats and reduce memory decline in an APP/PS1 mice model. Thus, this study aimed to assess the effect of ovocystatin on histopathological changes in APP/PS1 mice. Materials and methods: Ovocystatin was administered intraperitoneally for four weeks (40 μg/mouse) to 35-weeks-old transgenic (AD, n = 14) and wild type (NCAR, n = 15) mice (stock B6C3-Tg(APPswe, PSEN1dE9)85Dbo/Mmjax). A histopathological evaluation comprised antibodies directed against β-amyloid (1:400, SIG-39320-1000, Covance) and Tau (1:4000, AHB0042, Invitrogen). Three regions of the hippocampus— the dentate gyrus (DG) and the cornu ammonis (CA1 and CA3)—were analyzed by immunohistochemistry in each animal. All differences are expressed as percentage relative to the control group. Results: The main results showed that the percentage of immunoreactive area of β-amyloid, tau protein deposits in APP/PS1+ovCYS was decreased in DG, CA1, and CA3 regions compared with the APP/PS1 control, respectively (*p* < 0.05). Conclusions: Ovocystatin caused significant changes in the expression pattern of all investigated proteins in hippocampal tissues both in APP/PS1 and NCAR mice.

## 1. Introduction

Alzheimer’s disease (AD), as a progressive and neurodegenerative disorder, has substantial consequences for a patient’s quality of life and their carers. Heretofore, the effective treatment is still unknown and current treatments have been restricted only to cholinesterase inhibitors (rivastigmine, donepezil) and the antagonist of N-methyl-D aspartate (NMDA) receptor (memantine), which only affect the symptoms [1,2,3]. Moreover, the etiology of AD is associated with many factors, such as genetic, biological, or social ones [4]. It has been also suggested that inflammation plays an important role in the course of AD and bacterial, viral, or fungal infections might be crucial in the pathogenesis [5]. Indeed, recently published data [6] revealed that oral *Porphyromonas gingivalis* infection resulted in mice brain colonization and increased production of β-amyloid fragments 1–42 (Aβ_1–42_). Moreover, gingipains—the toxic proteases, are identified in AD patients’ brains and linked with protein tau and ubiquitin-related pathology. Nevertheless, β-amyloid (Aβ), apolipoprotein E (APOE), and protein tau are still considered the substantial elements which contribute to Alzheimer’s disease pathophysiology [3]. Several lines of evidence support that mutations in amyloid precursor protein (*APP*) and presenilin 1 (*PSEN1*) and 2 (*PSEN2*) genes lead to the development of the disease and production of toxic Aβ, especially in patients with early-onset autosomal dominant familial Alzheimer’s disease (FAD) [7,8]. From the neuropathological point of view, AD is characterized by amyloid plaques and neurofibrillary tangles (NFTs), followed by neurodegeneration with synaptic and neuronal loss causing macroscopic atrophy [9]. AD can be divided into three stages, according to the degree of cognitive impairment [10]. Memory loss, cognitive impairments, and behavioural changes are the main clinical manifestations of AD, affecting the daily activities of the affected individuals [11]. It has been evaluated that AD will affect 131 million people worldwide by 2050 and will cause over US $2 trillion in economic losses by 2030 [12]. Thus, novel therapeutic approaches concerning anti-amyloid therapy, anti-tau therapy, anti-neuroinflammatory therapy, neuroprotective agents including N-methyl-D-aspartate (NMDA) receptor modulators, and brain stimulation are still being searched [13]. Moreover, different tested bioactive compounds also have the potential for clinical application. Some proteins, carbohydrates, vitamins, fats, and oils may synadditively confer overwhelming protection against neurodegenerative diseases by modulating the activities of these critical enzymes of physiological importance [14].

Multiple lines of research have revealed that cysteine proteases play an important role in AD pathology [15]. It has been widely demonstrated that cysteine protease inhibitor-cystatin C (Cys C) might be a potential target for Alzheimer’s disease treatment [16]. Cys C has been found in all mammalian body fluids, such as cerebrospinal fluid (CSF), blood plasma, and all mononuclear cells. In the brain it is expressed by neurons, astrocytes, endothelial, and microglial cells (for review see: [17]). Cys C has a broad spectrum of biological functions ranging from modulation of inflammatory response, antibacterial and antiviral properties, to inhibition of tumor metastasis [18,19,20]. Interestingly, numerous studies have demonstrated that Cys C plays a crucial biological role in neurodegenerative disorders, especially in the pathophysiology of Alzheimer’s disease [21]. It has been shown that Cys C co-deposits with Aβ in Alzheimer’s disease patients’ brains [22] and the association between cystatin C and Aβ demonstrate a specific, saturable, and high-affinity binding between cystatin C and both Aβ_1–42_ and Aβ_1–40_ [23]. Additionally, in vitro studies reported inhibitory properties of Cys C against fibril formation and oligomerization [24]. Studies by Kaur and Levy [20] and Gauthier et al. [21] elaborate on the neuroprotective roles of Cys C, including the inhibition of cysteine proteases, such as cathepsins B, H, K, L, and S, the induction of autophagy, and the regulation of cell proliferation which is linked to the induction of neurogenesis. Similarly, oral administration of the cysteine protease inhibitor E64d improved memory deficit. Furthermore, reduced brain Aβ_40_ and Aβ_42_, amyloid plaque, and brain cathepsin B activity were observed [25].

In the light of these findings, a need of developing new therapeutic strategies comprising the prevention and treatment of Alzheimer’s disease has been highlighted. Thus, our study focused on ovocystatin (ovCYS), which is the best-characterized type 2 cystatin protein. Ovocystatin has been used in a series of experimental studies as a model protein representing the cystatin superfamily [26]. Likewise, cystatin C inhibits a broad range of lysosomal cathepsins, including cathepsin B, H, K, L, and S [27]. The protein is highly homologous to its human counterpart cystatin C, (62% structural similarity) and has similar biological properties [28]. In addition, it has been revealed that ovocystatin is characterized by relatively low immunogenicity [29]. Moreover, in contrast to cystatin C it can be easily obtained from chicken eggs on a large scale [30].

Recently, Stańczykiewicz and colleagues have shown that ovocystatin has beneficial properties for cognitive functions in young rats [31], and might prevent aging-related cognitive impairment in older animals [32] as well as reduce memory decline in the APP/PS1 mice model [33]. Moreover, it has been shown that six-months of ovocystatin administration in drinking water may become a safe, effective, and well-tolerated approach in the prevention of cognitive decline in APP/PS1 mice [33]. For these reasons, the protein seems to be a suitable tool for studying the role of cystatins in the pathophysiology of neurological disorders. Indeed, the prevention of neurodegeneration is an important aspect of modern medicine. Addressing several limitations in previous studies, related to a lack of morphological, biochemical, and immunohistochemical studies, our research can provide the first data of more reliable insights into the biological properties of ovocystatin. Heretofore, it has been shown that ovocystatin may inhibit the deterioration of cognitive functions without indicating crucial properties of ovocystatin, related to neurodegenerative processes. Hence, this study aimed to assess the effect of ovocystatin on histopathological changes in APP/PS1 mice.

## 2. Materials and Methods

### 2.1. Reagents—Isolation and Characterization of Ovocystatin

Egg white homogenate was applied on an affinity chromatography column containing S-carboxymethylated papain-Sepharose 4B. Ovocystatin was eluted with 50 mM K_3_PO_4_ containing 0.5 M NaCl at pH 11, dialysed against 50 mM NH_4_HCO_3_ and lyophilized. The anti-papain activity of the inhibitor was measured calorimetrically against α-N-Benzoyl-DL-arginine β-naphthylamide (BANA) as a substrate [34]. The purity of ovocystatin was checked by SDS-PAGE in 12% gel under reducing conditions [35]. Based on SDS-PAGE electrophoresis, the inhibitor was pure and not aggregated. A more detailed description of ovocystatin preparation was presented in another paper [33].

### 2.2. Animals—APP/PS1 Mice

Male APP/PS1 transgenic mice used for immunohistochemical analysis were purchased from Jackson Laboratory, Bar Harbor, ME, USA. These mice display the development of Aβ deposits by six months of age and express the mouse/human APPswe (K595N/M596L) and exon-9 deleted presenilin 1 (deltaE9) [36,37]. The mice were housed (three to four per cage) under standard conditions (12:12 light-dark cycle with lights on at 7:00 a.m.; temperature 21 ± 2 °C) and with free access to pelleted mouse chow and drinking water.

### 2.3. Animals Grouping, Intervention, and Sample Preparation

Thirty-five-week old APP/PS1 (B6C3-TG (APPswe, PSENdE9)85Dbo/J, initial body weight approximately 27 g; n = 14) and age-matched non-carrier (NCAR) control (initial body weight approximately 28.66 g; n = 15) were randomly assigned into control and treatment (ovCYS) groups, so at the end, there were four experimental groups: APP/PS1 (vehiculum, n = 7), APP/PS1 + ovCYS (n = 7), NCAR (vehiculum, n = 7), NCAR + ovCYS (n = 8). Mice were injected intraperitoneally for four weeks (five days every week with a two day gap). Ovocystatin was given at the dosage of 40 µg/mouse, corresponding to 100 µL of working solution, while control groups received vehicle injections (0.9% saline) starting at the age of 35-weeks. After four weeks of administration, mice were deeply anesthetized with a cocktail of ketamine and xylazine (50–100 and 5–10 mg/kg i.p. respectively) and perfused transcardially with 0.9% saline solution. Brains were dissected and cut into left and right hemispheres. The right hemisphere was fixed in 4% phosphate-buffered formalin and embedded in paraffin afterward.

All animal experiments were performed according to the National Institutes of Health Guide for the Care and Use of Laboratory Animals and were approved by the Local Ethical Committee. All efforts were made to minimize animal suffering and to reduce the number of animals used.

### 2.4. Immunohistochemistry

Immunohistochemical reactions were performed on 4-µm-thick paraffin sections using Autostainer Link48 (Dako, Glostrup, Denmark). Tissue sections were deparaffined, rehydrated and antigen retrieval was carried out by treating the slides with EnVision FLEX Target Retrieval Solution (97 °C, 20 min; pH 9) using a PT-Link. The activity of endogenous peroxidase was blocked by 5 min. incubation with EnVision FLEX Peroxidase-Blocking Reagent (Dako). Afterward, primary antibodies (diluted in EnVision FLEX Antibody Diluent (Dako)), β-amyloid (mouse Mo, anti-human; 1:400, SIG-39320-1000, Covance, Princeton, NJ, USA), and tau (mouse Mo, anti-human, mouse; 1:4000, AHB0042, Invitrogen, Waltham, MA, USA) were applied for 20 min. Next, slides were incubated with EnVision FLEX/ HRP (20 min). 3,3′-diaminobenzidine (DAB, Dako) was utilized as the peroxidase substrate and the sections were incubated for 10 min. Finally, all sections were counterstained with EnVision FLEX Hematoxylin (Dako) for 5 min. After dehydration in graded ethanol concentrations (70%, 96%, 99,8%) and xylene, slides were closed with coverslips in Dako Mounting Medium (Dako).

### 2.5. Image Analysis

The sections were evaluated under the BX-41 light microscope Olympus, Tokyo, Japan and were collected serially. For image analysis of immunohistochemistry, 3 coronal sections taken from the middle (bregma; −1.82 mm) hippocampus of each mouse was analyzed. All immunoreactive areas of plaque were quantitatively analyzed using ImageJ, version 1.49 (NIH, Bethesda, MD, USA). Quantification was performed under 40× magnification. Image processing included background correction, color deconvolution, and adjustment to obtain the true positive DAB signal. The average pixel density within the specific region of the hippocampus was then recorded for further calculations. The two different experimenters were blinded to genotype and treatment during image collecting and processing. Data are expressed as the percentage of the hippocampal region occupied by the positive DAB signal.

### 2.6. Statistical Analysis

Data were expressed as mean ± standard deviation and analyzed using IBM SPSS Statistics ver. 23 (IBM Corporation, Armonk, NY, USA). Two-way ANOVA with Bonferroni post-hoc comparisons was used to determine the differences between groups. Genotype/treatment and ROI were the independent variables and the percentage of the area with positive immunoreactivity was the dependent variable in two-way ANOVA. Statistical significance was set at *p* < 0.05.

## 3. Results

The effects of ovocystatin administration on amyloid plaque and pathological tau protein deposits were studied by immunohistochemical analysis of hippocampal slices in a mouse model of Alzheimer’s disease. The burden of β-amyloid and tau protein deposition in the hippocampus of transgenic (APP/PS1) and non-carrier (NCAR) mice treated with ovocystatin was analyzed. A quantitative analysis was repeated twice for precision. The average value of two measures was obtained in each animal. Three regions of the hippocampus (region of interest, ROI)—DG, CA1, and CA3—were analyzed in each animal. All differences are expressed as a percentage relative to the control group (Figure 1 and Figure 2).

The percentage of the positive immunoreactive area of β-amyloid (Figure 1a) in ovocystatin treated transgenic mice (APP/PS1 + ovCYS, n = 7) was decreased by 30.14 in DG (39.14 ± 3.00 vs. 69.29 ± 5.15, *p* < 0.001), 26.41 in CA1 (27.57 ± 1.69 vs. 53.98 ± 5.94, *p* < 0.001) and 26.14 in CA3 (44.71 ± 1.74 vs. 70.86 ± 5.59, *p <* 0.001), compared with the APP/PS1 control (n = 7), respectively. The percentage of the positive immunoreactive area of β-amyloid in ovocystatin treated non-carrier mice (NCAR + ovCYS, n = 8) was decreased by 15.51 in the dentate gyrus (13.25 ± 2.55 vs. 28.76 ± 3.61, *p <* 0.01), 10.14 in CA1 (13.25 ± 2.07 vs. 23.39 ± 2.67, *p >* 0.05) and 5.00 in CA3 (13.38 ± 2.88 vs. 18.38 ± 1.29, *p >* 0.05), compared with the NCAR control (n = 7), respectively. The strong statistically significant effect of treatment (F_(3,75)_ = 125.0, *p* < 0.0001) and ROI (F_(2,75)_ = 6.573, *p* = 0.0023), as well as the strong significant interaction between treatment and ROI (F_(6,75)_ = 3.059, *p* = 0.0099), was observed in the analysis. The interaction resulted from the difference (decrease) of the positive signal observed in the dentate gyrus, which was bigger in the ovCYS treatment groups.

The percentage of the positive immunoreactive area of tau protein deposits (Figure 1b) in the APP/PS1 + ovCYS group (n = 7) was decreased by 6.14 in the dentate gyrus (23 ± 0.62 vs. 29.14 ± 2.00, *p <* 0.05), 6.71 in CA1 (23.86 ± 0.46 vs. 30.57 ± 2.28, *p <* 0.01) and 5.71 in CA3 (26.29 ± 2.22 vs. 32 ± 3.06, *p <* 0.05), compared with the APP/PS1 control (n = 7), respectively. The percentage of the positive immunoreactive area of tau protein deposits in the NCAR + ovCYS group (n = 8) was decreased by 7.61 in the dentate gyrus (8.25 ± 0.65 vs. 15.86 ± 1.18, *p <* 0.01), 6.21 in CA1 (10.5 ± 0.68 vs. 16.71 ± 0.87, *p <* 0.05) and 8.82 in CA3 (8.75 ± 0.56 vs. 17.57 ± 0.92, *p <* 0.001), compared with the NCAR control (n = 7), respectively. The statistically significant effect of treatment (F_(3,75)_ = 120.3, *p* < 0.0001) was observed in the analysis.

## 4. Discussion

To our knowledge, this is the first study evaluating the histopathological changes in brain tissue of APP/PS1 mice after the intraperitoneal administration of ovocystatin. To date, only several studies have aimed to address alternations of cognitive functions after ovocystatin supplementation. Indeed, recently published data by Stańczykiewicz and colleagues [33] revealed that ovocystatin administered for six months in drinking water at a dose of 40 μg/mouse reduces memory deficits in APP/PS1 transgenic mice. Moreover, it was noted that ovocystatin was given orally and intraperitoneally to improve cognitive functions in young rats [31]. The potential protective effect of ovocystatin administered intraperitoneally on age-related cognitive impairments in rats was also determined by Stańczykiewicz et al. [32]. Nevertheless, the obtained results were not statistically significant, but highlight the potential role of ovocystatin in neurodegenerative disorders. Additionally, it has to be mentioned that prolonged ovocystatin administration did not affect physical activity and might be a safe and effective intervention for Alzheimer’s disease. However, the exact mechanisms of action by this mode of ovocystatin administration remain unclear.

We do not know whether or not intact ovocystatin had crossed the blood-brain barrier (BBB) and eventually reached the neuronal tissues. Perhaps ovocystatin might have reached the neuronal compartment by some mechanisms that are not well defined yet. The passage of proteins within extracellular vesicles from circulation to the brain and the other way out is well established and is implicated in the pathogenesis of neurodegenerative disorders [38,39,40]. It is also plausible that the protein did not reach the brain tissues but acted by peripheral interactions with the immune system. There is compelling evidence that apart from CNS inflammation, the peripheral immune response may be involved in the progression of neurodegenerative diseases [41].

Thus, for the first time, we performed the histopathological evaluation taking into account three regions of the hippocampus—the dentate gyrus (DG), CA1, and CA3. It comprised antibodies directed against β-amyloid and Tau, which reflects an apparent neuropathological change in the hippocampus—the intraneuronal accumulation of Aβ42 and microtubule stability, respectively. For example, in Alzheimer’s disease, tau accumulates in the somatodendritic compartment [42,43]. Interestingly, we found that ovocystatin ameliorates hippocampal neurodegenerative changes, and thus might be beneficial for Alzheimer’s disease treatment. Indeed, our findings showed that the percentage of positive immunoreactive areas of β-amyloid, tau protein deposits in APP/PS1 + ovCYS was decreased in DG, CA1, and CA3 regions compared with the APP/PS1 control. Moreover, the percentage of positive immunoreactive areas of β-amyloid and tau protein in ovocystatin treated NCAR was decreased in DG as well compared with the NCAR control. Hence, the obtained results imply protective mechanisms of ovocystatin in neurodegenerative diseases, which appears to be consistent with earlier findings for cystatin C biological functions (for review see [21]). We speculate that ovocystatin, due to its similarity to cystatin C [28], might induce protective pathways and prevent brain damage and neurodegeneration as well. There is convincing evidence that cystatin C plays an important role in aging and Alzheimer’s disease [17]. First of all, cystatin C co-deposits with Aβ [21], binds to both Aβ_1–42_ and Aβ_1–40_ and inhibits Aβ fibril formation [23].

Indeed, some in vivo reports revealed that cystatin C association with Aβ inhibits Aβ oligomerization [24,44]. Additionally, higher cystatin C expression diminishes Aβ deposition [45,46]. Secondly, it has been suggested that cystatin C is endocytosed by damaged neurons and targeted to the lysosome, which results in inhibition of some lysosomal proteases and protects the cell from excessive lysosomal activity dysfunction [47]. Moreover, Sun et al. [48] demonstrated that cystatin C can prevent the inhibitory mechanisms of Cathepsin B-mediated Aβ degradation. Tizon et al. [44] also demonstrated the direct protection of neuronal cells from Aβ toxicity and induced apoptotic cell death by cystatin C. Neuroinflammation is currently recognized as an important pathophysiological feature of AD [49]. Sustained activation of brain-resident microglia and other immune cells (also peripheral lymphocytes), has been shown to exacerbate both amyloid and tau pathology. It leads to the development of the permanent inflammatory process resulting in apoptosis of neurons in the brain. Hence, the ability to modulate the inflammatory response is an important therapeutic aspect of AD. Unfortunately, our in vitro studies using bone-marrow-derived macrophages BMDM and primary mouse microglia excluded the potential immunoregulatory effect of ovocystatin (unpublished data).

In Alzheimer’s disease, neuronal autophagy can also be impaired. This disfunction may block the neuroprotective effects of autophagy, promote neuronal cell death and apoptosis and lead to the accumulation of toxic proteins, such as the tau protein [50]. Cystatin C may restore full functional autophagy via the mTOR pathway, which can be crucial for cell adaptation and survival under extreme conditions [51]. It has to be noted that our findings are consistent with the above-mentioned result because we found not only a decrease in Aβ but also decreased tau protein levels in hippocampus areas. Nevertheless, a recently published study by Duan et al. [52] revealed that cystatin C inhibits turnover of GSK3β and promotes GSK3β-catalyzed tau phosphorylation. Intraneuronal lysosomal/autophagosomal pathology observed in APP/PS1 mice is believed to be connected to the proteolytic processing of APP and tau protein, leading to the generation of toxic carboxy-terminal fragments and oligomeric β-amyloid and truncated forms of tau protein [53]. Thus, compromising lysosomal proteolytic activity by cystatin (i.e., cathepsin B, L, H, and asparaginyl endopeptidase) may prevent the enhanced production of invalid APP and tau fragments. This is another indication that cystatin C plays a protective role via APP-stimulated increase in cystatin C secretion, which mediates neural stem/progenitor cells [54]. Moreover, cystatin C interaction with fibroblast growth factor 2 (FFG-2) could lead to neurogenesis stimulation in the dentate gyrus in the rat’s hippocampus. Regarding the current interesting reports about the role of *Porphyromonas gingivalis* in the course of Alzheimer’s disease [6], it should be noted that both cystatin C and ovocystatin have antibacterial activity, including *Porphyromonas ginigivalis* [55,56]. Hence, the exact role of cystatin C and ovocystatin in neuropathogenesis remains unclear and needs further studies.

However, this study has certain limitations that should be discussed. First, our sample was relatively small due to recommendations for using animals for research purposes. Second, the APP/PS1 transgenic mice are a model referring to the sporadic form of Alzheimer’s disease. Additionally, the effect of gender on the mechanisms of action, which is still not sufficiently explained, could not be assessed due to the fact that only male mice were enrolled in the study. Another point is that we applied ovocystatin only in one dose (40 μg/mouse) based on previously published studies by Stańczykiewicz et al. [33]. Moreover, the evaluation was only done after intraperitoneal administration, while the foregoing data suggest that oral supplementation has therapeutic properties. Hence, further studies using oral administration are warranted. It should also be noted that, similarly to our previous results [32,33] we have not assessed the influence of ovocystatin on the expression of endogenous mouse cystatin C [57]. Numerous lines of studies revealed that the inhibitor E64d is an excellent tool compound for preclinical testing [58], and significantly improves memory and reduces Aβ [59]. To the best of our knowledge, only two studies revealed the beneficial effect of in vivo cystatin C administration on the neurodegeneration process. Namely, Nagai and colleagues [60] revealed that β-amyloid was not deposited in the hippocampus following cystatin C administration, which could support our hypothesis that exogenous ovocystatin inhibits β-amyloid oligomerization and deposition and may reduce cognitive deterioration, as we previously discussed [36]. Moreover, recently published results by Watanabe et al. (2018) showed that intraventricular administration of Cys C demonstrates a neuroprotective effect in other neurodegenerative diseases, i.e., amyothropic lateral sclerosis (ALS), and suggest that Cys C may represent a novel therapeutic candidate for ALS [61]. Taken together, all these results indicate that cystatin C and other inhibitors such as E64 reduce β-amyloid and might play a pivotal role in neuroprotection. Thus, using other inhibitors as controls, such as cystatin C and general inhibitors of cysteine peptidases E64, would significantly improve the value of our results. These limitations should be addressed in further studies on ovocystatin’s mechanism of action in the CNS.

## 5. Conclusions

Findings from our study point to the hypothesis that the administration of ovocystatin may not only reduce memory impairment in APP/PS1 transgenic mice, as was shown previously, but might also be a useful agent against Aβ oligomerization and consequent amyloid fibril formation and tau protein deposition. However, to confirm this hypothesis, more morphological, biochemical, and immunohistochemical analyses are needed. Further pharmacokinetic, stability, and distribution studies are necessary to assess its potential therapeutic properties.

## Figures and Tables

**Figure 1 jcm-11-02372-f001:**
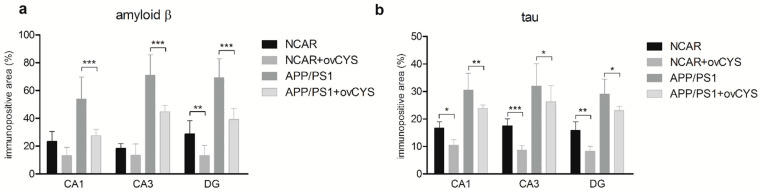
Immunoreactive area (percentage of the total area measured) of β-amyloid (**a**) and tau (**b**) burden in three hippocampal regions. (**a**) the burden of β-amyloid significantly decreased in the ovocystatin-treated transgenic group within all analyzed regions of the hippocampus, (**b**) deposits of misfolded tau protein significantly decreased in both non-carrier and transgenic ovocystatin-treated groups within all analysed hippocampal regions. NCAR—non-carrier group, NCAR+ovCYS—ovocystatin-treated non-carrier group, APP/PS1—transgenic group, APP/PS1+ovCYS—ovocystatin-treated transgenic group. Statistical significance of treated ovCYS versus untreated groups was indicated as * at *p* < 0.05, ** at *p* < 0.01, and *** at *p* < 0.001 within the respective hippocampal region.

**Figure 2 jcm-11-02372-f002:**
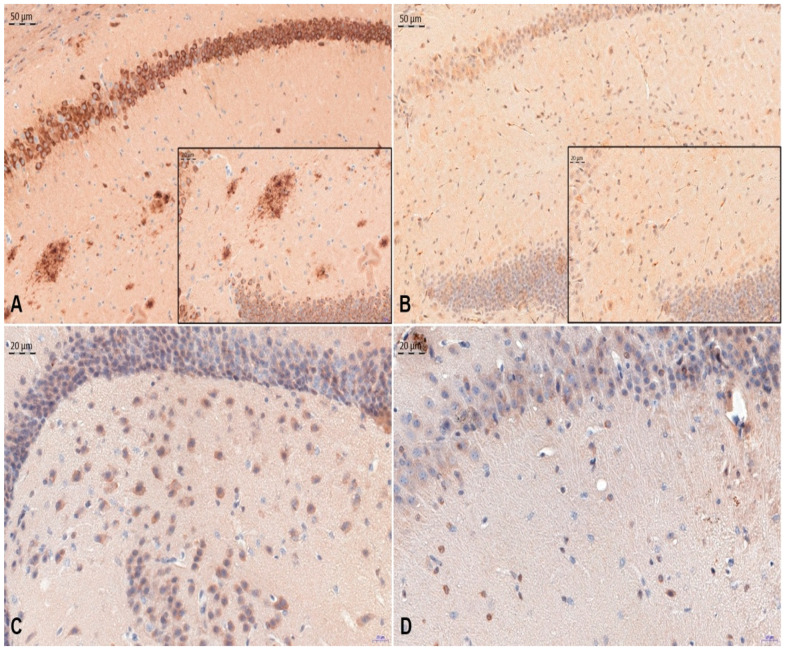
Representative images of immunohistochemical reactions indicating β-amyloid (**A**,**B**) and TAU-5 (**C**,**D**) antigen expression were carried out on APP/PS1 + ovCYS group (**A**,**C**) and NCAR control (**B**,**D**) mouse brain. Nuclei are stained using hematoxylin. Magnification ×200 (**A**,**B**) and ×400 (**C**,**D** and insert).

## Data Availability

The data analyzed during this study are included in this published article. Further inquiries can be directed to the corresponding authors.

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
