# Peer review of "Ovocystatin Induced Changes in Expression of Alzheimer’s Disease Relevant Proteins in APP/PS1 Transgenic Mice"

_jcm, 2022, doi:10.3390/jcm11092372_

Round 1

Reviewer 1 Report

Here are some of my suggestions for this article:
-Please check
the acronyms used in the text. Once you have defined an acronym, it should be used throughout the text. For example for words Alzheimer's disease (L76, L165, L220, etc); ovocystatin (L85, L176, L223, etc); beta-amyloid (L27, L31, L172,etc), etc.
-Please check the sentence style (L133 and 134).
-Define in the text the acronym BBM.
-I suggest changing the text in Figure 2, indicating that it is a representative image.

Author Response

Point 1: -Please check the acronyms used in the text. Once you have defined an acronym, it should be used throughout the text. For example for words Alzheimer's disease (L76, L165, L220, etc); ovocystatin (L85, L176, L223, etc); beta-amyloid (L27, L31, L172,etc), etc.

Thank you for commenting this issue. All acronyms have been unified.

Point 2: Please check the sentence style (L133 and 134).  

Thank you that you raised this issue. All manuscript has been checked by English native speaker.

Point 3: Define in the text the acronym BBM.

In the first version of the manuscript, we used the term BBM for blood brain membrane. Nevertheless, the term blood-brain barrier is used more frequently in the literature. Hence, we decided to change BBM to blood brain barrier with the acronym BBB.

Point 4: I suggest changing the text in Figure 2, indicating that it is a representative image.

We have added this suggestion to the Figure 2’ description.

Reviewer 2 Report

Stanczykiewicz et al., entitled manuscript “Ovocystatin Induced Changes in Expression of Alzheimer’s Disease Relevant Proteins in APP/PS1 Transgenic Mice,” aimed to investigate the potential neuroprotective function of Ovocystatin in APP/PS1 Transgenic Mice models. Here, the authors also aimed to investigate the effect of ovocystatin induced changes in expression of Alzheimer’s disease relevant proteins. However, the authors lack the command over the topic despite having their interest over the topic. Here, the reviewer mentioned some critical issues, which should be clarified before publishing the article to increase the scientific relevance of the study.

  1. It is well established fact that depression-like behavior is linked with Alzheimer’s disease. The authors are suggested to perform behavioural test, namely locomotor activity recording, Morris water maze, and forced swim test as this will demonstrate the role of ovocystatin administration on behaviour of the APP/PS1 mice. The authors should include an additional figure demonstrating the effects of ovocystatin on the acquisition training in the MWM test and the influence of administering ovocystatin on locomotor activity. Further, the authors are suggested to perform cognitive assessment test, namely elevated plus maze, Y maze, and object recognition test.
  2. The authors should test the effect of ovocystatin on serum T3 and T4 levels. Oxidative stress and neuroinflammation are considered as important pathogenic mechanism in the pathogenesis of Alzheimer’s disease. However, the authors failed to mention about oxidative stress and neuroinflammation. Thus, the authors are suggested to evaluate the expression of biomarkers of neuroinflammation, namely Il-1β, TNF-α, and microglial proteins, such as IbA1 and GFAP through western blot analysis or ELISA.
  3. Moreover, the authors are suggested to evaluate the effect of ovocystatin administration of biomarkers of oxidative stress, such as MDA, GSH-Px, and SOD. Further, the authors should demonstrate the effect of ovocystatin treatment on atrophy in the CA1 hippocampus of APP/PS1 mice.
  4. Neuronal apoptosis is another characteristic feature of AD pathogenesis. However, the authors did not mention about the expression change of apoptotic proteins, namely Bcl2, Bax, Bak, caspase-3, and caspase-9 before and after ovocystatin treatment.
  5. In addition, neurotrophins, namely BDNF, NGF, NT-3, and NT-4 were play an important role in the pathogenesis of Alzheimer’s disease. Thus, the authors are suggested to evaluate the expression changes before and after ovocystatin treatment.
  6. Nowadays, many drugs are being used for repurposing. Moreover, many natural compounds are also being taken to investigate how they would improve the symptoms of Alzheimer’s disease. But here the author took ovocystatin which is being extracted from chicken egg white. It would be interesting to know what were the inclusion and exclusion parameters behind taking ovocystatin. Also, besides curing memory impairment and slowing down the progression of cognitive decline, what other prospects does ovocystatin hold? Authors can add some future perspectives of the same.
  7. Although there is a strong relationship between, cystatin C and APOE4. Studies done in past suggest that CST 3 and APOE4 indicated a higher fold of amyloid-beta plaques. However, here author suggests a neuroprotective effect of cystatin. But it would be enthralling if authors can put some light upon the working combination of ovocystatin and APOE4.
  8. Does body weight have an influence on dosages of ovocystatin? Or was it one dose for all? Moreover, what will be the possible outcome for overdosing of ovocystatin?
  9. The authors utilized only semi-quantitative approach which is Immunohistochemistry experiment to calculate the expression analysis. It is recommended that the authors back up their findings with western blot analysis or ELISA of protein isolated from tissue samples.
  10. Authors are advised to provide “scale bar” in IHC image used in the manuscript.
  11. For oral delivery, the authors employed a dosage concentration of Ovocystatin of 40 g/mouse. The use of this precise dose of Ovocystatin should be justified by the authors. Data may be supported by authors.
  12. Authors are requested to mention regarding the effect of Ovocystatin administration on oxidative stress and neuroinflammation is encouraged, as these are significant mechanisms linked to Alzheimer's disease.
  13. The authors are required to report the toxicity profile of the APP/PS1 Transgenic Mice model after Ovocystatin administration (such as chronic oral toxicity tests, neurotoxicity, nephrotoxicity, and so on). This information will reveal whether or not the concentration is harmful to animal health.
  14. Authors are advised to briefly explain the reason for using only Male mice for experiments.
  15. The effect of Ovocystatin on neuronal death or brain shrinkage should be investigated by the authors. Many studies have found that neuronal loss occurs in Alzheimer's disease.Hence it is important to provide data in this context. For reference: https://doi.org/10.1016/j.isci.2021.102942
  16. Moreover, authors are advised to provide the data about how administration of ovocystatin effects behavior by performing behavioral tests such as The Morris Water Maze, Radial Arm Maze. These experiments can be performed before sacrificing mice for sample collection.  For reference: doi: 10.1016/j.bcp.2014.01.011.

Author Response

Reviewer suggestion 1.  It is well established fact that depression-like behavior is linked with Alzheimer’s disease. The authors are suggested to perform behavioral test, namely locomotor activity recording, Morris water maze, and forced swim test as this will demonstrate the role of ovocystatin administration on behavior of the APP/PS1 mice. The authors should include an additional figure demonstrating the effects of ovocystatin on the acquisition training in the MWM test and the influence of administering ovocystatin on locomotor activity. Further, the authors are suggested to perform cognitive assessment test, namely elevated plus maze, Y maze, and object recognition test.

Answer to suggestion 1. Thank you for these comments. Indeed, we conducted the behavioral test (the locomotor activity and cognitive functions were determined using an actimeter and the Morris water maze test, respectively). Nevertheless, the ovocystatin’ administration do not influence on neither travelled distance nor permanence time in the SW target zone (APP/PS1 and NCAR groups), where the hidden platform was previously located, during the probe trial (p>0.05). However, untreated APP/PS1 mice spent less time, and travelled less distance in the SW target zone than untreated NCAR mice, but the differences were not statistically significant (APP/PS1 vs. NCAR, p>0.05). This can imply slight cognitive impairments caused by genetic modification. Probably, these slight changes might be caused by very short administration. The number of animals was also not inconsiderable as we mentioned at discussion. Thus, we decided to focus only at the changes obtained in the staining.

Reviewer suggestion 2. The authors should test the effect of ovocystatin on serum T3 and T4 levels. Oxidative stress and neuroinflammation are considered as important pathogenic mechanism in the pathogenesis of Alzheimer’s disease. However, the authors failed to mention about oxidative stress and neuroinflammation. Thus, the authors are suggested to evaluate the expression of biomarkers of neuroinflammation, namely Il-1β, TNF-α, and microglial proteins, such as IbA1 and GFAP through western blot analysis or ELISA.

Answer to suggestion 2. Thank you for this valuable comment. In another study of our group the immunoregulatory activity of ovocystatin was determined using two different cellular models: mice bone marrow derived macrophages of BMDM cell line primary mouse microglia. The impact of ovocystatin to produce inflammatory mediators was studied. Firstly, it was checked that ovocystatin at doses ranging from 1 to 150 μg/ml is not toxic to these cells (MTT test and SRB test). It has been shown no effect of ovocystatin on production and secretion of nitric oxide (NO) and cytokines: proinflammatory TNF alpha and anti-inflammatory IL-10 (unpublished data). Unfortunately, we have currently no possibility to test the serum T3 and T4 levels, because of lack of research material (serum). However, thank you for your comment, which we will include in our future research plans.

Reviewer suggestion 3.  Moreover, the authors are suggested to evaluate the effect of ovocystatin administration of biomarkers of oxidative stress, such as MDA, GSH-Px, and SOD. Further, the authors should demonstrate the effect of ovocystatin treatment on atrophy in the CA1 hippocampus of APP/PS1 mice.

Answer to suggestion 3. Thank you for this important remark. Tests on the influence of ovocystatin will be performed in the near future on cell lines. These experiments require numerous repetitions, very precise methods and stable research material, especially with regard to the analysis of GSH-Px and SOD activity.

Reviewer suggestion 4. Neuronal apoptosis is another characteristic feature of AD pathogenesis. However, the authors did not mention about the expression change of apoptotic proteins, namely Bcl2, Bax, Bak, caspase-3, and caspase-9 before and after ovocystatin treatment.

Answer to suggestion 4. We are currently carrying out the research project entitled “The impact of ovocystatin on neurodegenerative processes – in quest of mechanisms participating in neurogenesis, and neuroprotection”. One of the research tasks try to explain the impact of ovocystatin on the regulation of the amyloid beta 42-induced apoptosis in primary hippocampal neurons H19-7. The tests will include the determination of cytotoxicity, activation of caspases 3/7, activation of ERK 1/2, JNK and p38 kinases, activation of the NF-kB factor and the expression of pro and anti-apoptotic proteins (Bcl2, Bax). We plan to publish the obtained results (due to their extensive scope) as a separate work. The research will start at the end of May 2022.

Reviewer suggestion 5.  In addition, neurotrophins, namely BDNF, NGF, NT-3, and NT-4 were play an important role in the pathogenesis of Alzheimer’s disease. Thus, the authors are suggested to evaluate the expression changes before and after ovocystatin treatment.

Answer to suggestion 5. Thank you for this suggestion. In our ongoing project, as we mentioned above, the assessment of the ovocystatin’ effect on the process of neurogenesis, neuritogenesis and BDNF production is conducted. The influence of ovocystatin on the survival of primary H19-7 neurons, their proliferation, neurogenesis and neuritogenesis were determined.  There was no effect of ovocystatin on neuritogenesis and BDNF production. However, the significant increase of neurogenesis marker was observed. This experiment is still ongoing.

Reviewer suggestion 6. Point Nowadays, many drugs are being used for repurposing. Moreover, many natural compounds are also being taken to investigate how they would improve the symptoms of Alzheimer’s disease. But here the author took ovocystatin which is being extracted from chicken egg white. It would be interesting to know what were the inclusion and exclusion parameters behind taking ovocystatin. Also, besides curing memory impairment and slowing down the progression of cognitive decline, what other prospects does ovocystatin hold? Authors can add some future perspectives of the same.

Answer to suggestion 6. Ovocystatin can be obtained in big amounts from eggs, which are easily extendible natural material as compared to animal or plant tissue resources. Moreover, the sequence and structure similarity between human and chicken cystatin make it low immunogenic. It has been revealed neuroprotective roles of Cys C, including inhibition of cysteine proteases, such as cathepsins B, H, K, L, and S, induction of autophagy, and regulation of cell proliferation which linked to induction of neurogenesis. Furthermore, reduced brain Aβ40 and Aβ42, amyloid plaque, and brain cathepsin B activity was observed. Based on these findings we assumed that ovocystatin may be a good model protein for the study with similar properties which can be evaluated. This has been shortly mentioned in the Introduction section.

Reviewer suggestion 7. Although there is a strong relationship between, cystatin C and APOE4. Studies done in past suggest that CST 3 and APOE4 indicated a higher fold of amyloid-beta plaques. However, here author suggests a neuroprotective effect of cystatin. But it would be enthralling if authors can put some light upon the working combination of ovocystatin and APOE4.

Answer to suggestion 7. Thank you for your suggestion highlighting this interesting issue. This is a very important remark that we have to take into account planning our further research.

A role for CST 3 in the pathogenesis of Alzheimer’s disease has been suggested by the genetic linkage of a CST 3 gene polymorphism with AD and the co-localization of CST 3 with amyloid beta, and ability of CST 3 to inhibit amyloid fibrillation. There are also some data indicating that the presence of CST 3 A allele (AG or AA genotypes) significantly increases the risk for APOE4 in AD patients [Cathcart et al., Neurology 2005, 64:757; Hua et al., 2012]. However, CST 3 A is not a risk factor for subjects lacking the APOE4 allele as ϵ24, ϵ34 or ϵ44 genotype.  Nevertheless, the research group of Maruyama’s (Neurology, 2001, 57, 337) or Dodel’s (Neurology, 2002, 58, 664) did not confirm these observations. Also, Lin et al. observed that either CST 3 or its interaction with APOE4 were not significant predictors of AD [Chin J Physiol., 2003, 46: 111]. Despite, the research into the role of human CST 3 in AD is very controversial, it is debatable point to be taken into account in our future study.  

The association between amyloid beta and CysC prevented amyloid beta accumulation and fibrillogenesis, indicating that CysC plays a protective role in the pathogenesis of AD in humans and explains why decreases in CysC concentration caused by the CST3 polymorphism can lead to the development of the AD disease. However, we would like to point out that studies described above concern endogenous human cystatin C expressed intracellularly, with the potential risk of genetic mutations. On the other hand, our research concerns exogenously applied ovocystatin, what eliminates this potential risk of gene mutations. 

However, we would like to emphasize that our latest results support the thesis about the neuroprotective effect of ovocystatin in AD. It was shown in previous reports the association between amyloid beta and Cys C in prevention of amyloid accumulation and fibrillogenesis. The inhibitory effects in vivo in amyloid beta-depositing transgenic amyloid-beta protein precursor mice overexpressing human CysC was shown. Also, a reduction in amyloid beta load was observed in the AβPP/CysC double transgenic mice compared to single AβPP transgenic mice [Kaeser et al., Nat Genet. 2007;39:1437–1439; Mi et al., Nat Genet. 2007;39:1440]. In our latest studies, the significant effect of ovocystatin isolated from egg white on the inhibition of amyloid beta 42 aggregation was determined (unpublished). ThT fluorimetric assay, TEM and CD spectroscopy were used in these studies. The manuscript is presently prepared to submit. These results point to the one of the possible neuroprotective mechanisms of action of ovocystatin.

Reviewer suggestion 8. Does body weight have an influence on dosages of ovocystatin? Or was it one dose for all? Moreover, what will be the possible outcome for overdosing of ovocystatin?

Answer to suggestion 8. All mice included in the experiments have similar weight 27+/-2 g as it was mentioned in the Materials and Methods section. Moreover, the body weight did not statistically change during the experiment period. Thus, the only one dose was applied. We do not know the possible effects of overdosing. The dose of ovocystatin was chosen based on the authors’ preliminary data (unpublished data) and previous study with the oral administration of ovocystatin. Thus, we indicated this issue in the limitations of the study. Further studies using different dosages of ovocystatin are warranted. More information concerning the dosage is given in answer to point 11.

Reviewer suggestion 9. The authors utilized only semi-quantitative approach which is Immunohistochemistry experiment to calculate the expression analysis. It is recommended that the authors back up their findings with western blot analysis or ELISA of protein isolated from tissue samples.

Answer to suggestion 9. Thank you for this important remark. We agree that verification of our results by other independent methods would be the advantage. However, this has not been included in this experimental protocol. We will introduce ELISA or another relevant test in our further studies.

Reviewer suggestion 10. Authors are advised to provide “scale bar” in IHC image used in the manuscript.

Answer to suggestion 10. The quality of the scale bars in the IHC images has been improved.

Reviewer suggestion 11. For oral delivery, the authors employed a dosage concentration of Ovocystatin of 40 g/mouse. The use of this precise dose of Ovocystatin should be justified by the authors. Data may be supported by authors.

Answer to suggestion 11. The effect of ovocystatin on immunocytotoxicity was investigated in SRBC (sheep red blood cells) immunized mice. Animals received tested substance 4 times at doses of 2, 0.2 and 0.02mg/kg, intraperitoneally. SRBC immunization was performed 24 hours after the last dose, or 2 hours before the first dose of the tested compound. The number of antibodies forming cells (AFC), total and 2-mercaptoethanol resistant serum agglutination titers were determined.

In in vitro studies, the impact of ovocystatin on viability of murine bone-marrow derived macrophages BMDM, primary mouse microglia and primary hippocampal neurons of H19-7 cell line was also tested using MTT and SRB tests. It was checked that ovocystatin at doses ranging from 1 to 150 μg/ml is not toxic to these cells. 

Additionally, cytotoxicity was examined on murine macrophage cell line J774.E, originally isolated from BALB/cN mice ascites and D10.G4.1 murine lymphoblasts. We did not observe immunocytotoxic effects of the protein at tested doses ranging from 1 – 150 ug/ml. Higher doses than 2 mg/ml were not examined because of the instability of ovocystatin in higher concentrations (aggregation).

Reviewer suggestion 12. Authors are requested to mention regarding the effect of Ovocystatin administration on oxidative stress and neuroinflammation is encouraged, as these are significant mechanisms linked to Alzheimer's disease.

Answer to suggestion 12. Thank you that you raised this important issue. As we pointed in answer to suggestion 2, the immunoregulatory activity of ovocystatin was determined using in vitro models. However, no regulatory of ovocystatin on the secretion of inflammatory factors by macrophages of the BMDM cell line and mouse microglia was observed. Because the neuroinflammation and oxidative stress are very important in Alzheimer’s disease pathogenesis, the fragment below, explaining the influence of ovocystatin on the above processes, has been added to the discussion: “Neuroinflammation is currently recognized as an important pathophysiological feature of AD. Sustained activation of brain-resident microglia and other immune cells (also peripheral lymphocytes), has been shown to exacerbate both amyloid and tau pathology.  It leads to the development of the permanent inflammatory response resulting in apoptosis of neurons in the brain. Hence, the ability to modulate the inflammatory response is an important therapeutic aspect in AD. Unfortunately, our in vitro studies using bone-marrow derived macrophages BMDM and primary mouse microglia excluded the potential immunoregulatory effect of ovocystatin (unpublished data).”

Reviewer suggestion 13. The authors are required to report the toxicity profile of the APP/PS1 Transgenic Mice model after Ovocystatin administration (such as chronic oral toxicity tests, neurotoxicity, nephrotoxicity, and so on). This information will reveal whether or not the concentration is harmful to animal health.

Answer to suggestion 13. Thank you for your suggestion highlighting this interesting remark. The preliminary toxicity tests have been performed before the administration (vide point 11 and 12). According to these data, limited number of mice and their cost, we also decided which dose was used in our study. As it was mentioned in the manuscript we observed in previous study (Stańczykiewicz et al. Advances in Medical Science, 2016: 64:65-71) that prolonged ovocystatin administration (6 months) did not affect the physical activity, and do not influence on the animal health and mortality of the mice. We also did not observe any health problems during this 2-weeks intraperitoneally administration in this study.

Reviewer suggestion 14. Authors are advised to briefly explain the reason for using only Male mice for experiments.

Answer to suggestion 14. Thank you for this comment. In this study we decided to conduct the experiments only on male mice to avoid any impact of endocrine changes occurring in female subjects. We agree that assessment of the gender’ influence on the cognitive behaviour of mice or other relevant changes after administering ovocystatin will be valuable.

We raised this issue in the limitations of the study.

Reviewer suggestion 15. The effect of Ovocystatin on neuronal death or brain shrinkage should be investigated by the authors. Many studies have found that neuronal loss occurs in Alzheimer's disease. Hence it is important to provide data in this context. For reference: https://doi.org/10.1016/j.isci.2021.102942

Answer to suggestion 15. We are fully agreed that assessment of the ovocystatin’ influence on neuronal death and brain shrinkage is crucial. Heretofore, the effect of ovocystatin on the survival of primary hippocampal neurons H19-7, neuron-like PC12 cells, BMDM cells and primary mouse microglia was determined in vitro. Viability was checked using ovocystatin at doses ranging from 1 to 150 ug/ml. These doses were not toxic to these cells (MTT test and SRB test). Additionally, no effect of ovocystatin on neuritogenesis of H19-7 and PC12 cells was shown. However, we observed significant increase in neurogenesis markers in H19-7 cells (Western blotting). Currently, the impact of ovocystatin on regulation of molecular signaling responsible for survival, proliferation and neuronal markers expression is studied.

Reviewer suggestion 16. Moreover, authors are advised to provide the data about how administration of ovocystatin effects behavior by performing behavioral tests such as The Morris Water Maze, Radial Arm Maze. These experiments can be performed before sacrificing mice for sample collection.  For reference: doi: 10.1016/j.bcp.2014.01.011.

Answer to suggestion 16. Thank you for this comment. Indeed, animal behavior evaluation is a fundamental tool in multiple areas of translational neuroscience and is useful for studying the efficacy of novel drugs in reversing phenotypes in disease models. We previously reported the influence of the ovocystatin administration in the intraperitoneally and oral way in the rats and APP/PS1 model as well (Stańczykiewicz B, et al. Adv Clin Exp Med. 2017; Postep Hig Med Dosw [Internet]. 2017; Adv Med Sci. 2019). We also performed behavioral tests during this study. The reason why we do not present these data in the manuscript was explained in the answer 1.

Reviewer 3 Report

1.      Authors are requested to critically check and edit their manuscript through a native English editor which will result in improving the quality of your manuscript.

2.      Authors should explain the background of the study in such a way that it may become easy for the readers to understand the background of the study easily.

For example, the authors should start the introduction from Alzheimer’s disease, giving little details about the disease, its signs, and symptoms, research done so far related to the study, and its outcomes. This will help and provide a better understanding for the readers.  

3.      Authors are advised to summarize the findings of previous studies at the end of the introduction, mentioning the importance of this study and its novelty as compared to previous studies.  

Author Response

Point 1.      Authors are requested to critically check and edit their manuscript through
a native English editor which will result in improving the quality of your manuscript.

Thank you that you raised this issue. All manuscript has been checked by English native speaker.

Point 2.      Authors should explain the background of the study in such a way that it may become easy for the readers to understand the background of the study easily.

For example, the authors should start the introduction from Alzheimer’s disease, giving little details about the disease, its signs, and symptoms, research done so far related to the study, and its outcomes. This will help and provide a better understanding for the readers.  

Thank you for your comment about the Introduction section and suggesting valuable improvement of our paper. We changed the section accordingly.

Point 3.      Authors are advised to summarize the findings of previous studies at the end of the introduction, mentioning the importance of this study and its novelty as compared to previous studies.  

Thank you for this important remark. The summary was added in the end of the Introduction.

Round 2

Reviewer 2 Report

The manuscript is now significantly improved for the publication in the current form.